# Strategic Development of an Immunotoxin for the Treatment of Glioblastoma and Other Tumours Expressing the Calcitonin Receptor

**DOI:** 10.3390/cells10092347

**Published:** 2021-09-08

**Authors:** Pragya Gupta, David L. Hare, Peter J. Wookey

**Affiliations:** Department of Medicine-Austin, University of Melbourne, Heidelberg 3084, Australia; pragyag@student.unimelb.edu.au (P.G.); david.hare@unimelb.edu.au (D.L.H.)

**Keywords:** brain tumour, glioblastoma, glioma stem cells, antibody, calcitonin receptor, CT Receptor, G protein-coupled receptor

## Abstract

New strategies aimed at treatment of glioblastoma are frequently proposed to overcome poor prognosis. Recently, research has focused on glioma stem cells (GSCs), some quiescent, which drive expansion of glioblastoma and provide the complexity and heterogeneity of the tumour hierarchy. Targeting quiescent GSCs is beyond the capability of conventional drugs such as temozolomide. Here, we discuss the proposal that the calcitonin receptor (CT Receptor), expressed in 76–86% of patient biopsies, is expressed by both malignant glioma cells and GSCs. Forty-two percent (42%) of high-grade glioma (HGG; representative of GSCs) cell lines available from one source express CT Receptor protein in cell culture. The pharmacological calcitonin (CT)-response profiles of four of the HGG cell lines were reported, suggesting mutational/splicing inactivation. Alternative splicing, commonly associated with cancer cells, could result in the predominant expression of the insert-positive isoform and explain the atypical pharmacology exhibited by CT non-responders. A role for the CT Receptor as a putative tumour suppressor and/or oncoprotein is discussed. Both CT responders and non-responders were sensitive to immunotoxins based on an anti-CT Receptor antibody conjugated to ribosomal-inactivating proteins. Sensitivity was increased by several logs with the triterpene glycoside SO1861, an endosomal escape enhancer. Under these conditions, the immunotoxins were 250–300 times more potent than an equivalent antibody conjugated with monomethyl auristatin E. Further refinements for improving the penetration of solid tumours are discussed. With this knowledge, a potential strategy for effective targeting of CSCs expressing this receptor is proposed for the treatment of GBM.

## 1. Introduction

Understanding the complexity and hierarchy of solid tumours, as well as cell constituents, is important background knowledge for the formulation of successful treatment strategies. In this minireview, the focus will be on the brain tumour glioblastoma to provide an example of the complexity of solid tumours as well as progress towards defining strategies for successful therapies, which optimally, would result in total tumour regression without minimal residual disease.

Glioblastoma (multiforme, GBM) is the most common malignant primary brain tumour in adults. The mean overall survival achieved with wide-local surgical resection, followed by adjuvant therapy of radiation and temozolomide (TMZ) plus maintenance therapy with TMZ, remains only 14.6 months. GBMs are highly heterogeneous tumours [1] thought to be derived from oligodendrocyte-type 2-astrocyte (O-2A) progenitors [2] of the astroglial lineage [3]. Furthermore, while GBMs can appear histologically similar, there are clear subtypes separable based on the lineage of the initiating cells [4].

## 2. Tumour Hierarchy of Glioblastoma

The tumour hierarchy or architecture includes regions of necrosis and oedema [5], where the blood–brain barrier is compromised and antibodies are able to penetrate [6] pro-inflammatory micro-environments with cellular infiltrates and vascular-like structures (vascular mimicry [7]) with proliferative domains, which are located primarily towards the tumour periphery. Within these latter hyperplastic structures, pericyte precursors and endothelial cells share the same genetic modifications as glioma stem-like cells (GSC), supporting plasticity with regard to differentiation lineages and the existence of dominant GSC clones [8]. It is noteworthy that pericyte precursors are pluripotent [9], are present in the structures of vascular mimicry and GBM vasculature, and in the close vicinity to the zones of proliferation, which raises the possibility that re-emerged GBM post-treatment might be derived from the pericyte lineage.

Figure 1A provides a schematic showing the salient cellular features of GBM while in Figure 1B regions of oedema (Oed) and structures of vascular mimicry (VM) are indicated in a patient biopsy.

The current inability to improve or predict patient outcomes based on genetic profiling or histopathological features points to a deficit in our understanding of the driving forces for tumourogenesis of GBM before and after therapy, and hence viable targets for tumour reduction or potential therapy. Currently, there are compelling data from animal models that GSCs play roles in tumourogenesis, tumour expansion, and re-establishment of the tumour hierarchy following treatments [10]. Small populations of GSCs have been identified that are quiescent and are resistant to conventional therapies that target dividing cells. These GSCs therefore offer a new potential target for therapies.

## 3. Cancer Stem Cells

The concept of cancer stem cells (CSCs) dates back to the early 1960s [11], however strong evidence was forthcoming from studies with leukaemic stem cells [12,13,14]. Transplantation of primary acute myeloid leukaemia (AML) cells into SCID [12] or NOD/SCID [13] mice (SCID: severe combined immune-deficient and NOD/SCID: non-obese diabetic/SCID) led to the finding that only rare cells, termed SCID leukaemia-initiating cells (SL-ICs), are capable of initiating and sustaining growth of the leukemic clone in vivo. In addition to their ability to differentiate and proliferate, serial transplantation experiments showed that SL-ICs possess the capacity for self-renewal, and thus can be considered as AML stem cells. Quiescent LSCs were subsequently shown to account for minimal residual disease [15].

This concept of CSCs was extended to breast cancer [16,17] and brain tumours [18,19]. Several studies have provided data in support. Brain tumour-initiating cells were enriched in experiments using the marker CD133 [18,19], and CD133^+^/CT Receptor^+^ cells were identified in GBM biopsies from patients [20]. In further studies, in models of medulloblastoma, CD15 was identified as a marker of tumour-propagating cells [21,22]. Rare quiescent sox2(+) cells, which are enriched following anti-mitotic treatment (temozolomide, TMZ), drive tumour regrowth in a similar mouse model of medulloblastoma [23]. In this context, conventional treatments of GBM following cyto-reductive surgery, namely, radiotherapy and chemotherapy, which target proliferating cells, will likely have a low probability of success. Furthermore, a platelet-derived growth factor (PDGF)-driven glioma subpopulation of tumour cells has been characterised as a stem-like population with the preference for the perivascular niche [24], which perhaps suggests their identity as pericyte precursors.

GSCs display inherent functional diversity [1,7,25], convey relative resistance to conventional treatments such as chemo- and radio-therapy [26], and provide invasive potential [19,27,28]. GSCs may also contribute to tumour survival and expansion in a number of hostile (hypoxic, inflammatory) micro-environments. As found with normal stem cell populations, GSCs are likely to be associated with dense vascular beds [29], which are generally located towards the periphery of GBM, and GSCs are believed to be present within the surrounding neuropil, perhaps in a quiescent state.

Further discussion about the identity of brain tumour stem cells has been reviewed recently [30].

## 4. Quiescent Brain Tumour Stem Cells

Accounting for functional heterogeneity within tumours and resistance to therapies remain challenging prospects for treatment strategies. Chen et al. [10] describe elegant mouse experiments which demonstrate the presence and responses of quiescent stem cells following TMZ treatment in a model of GBM. It was found that a subset of these endogenous quiescent cells share properties with GSCs and are resistant to TMZ (most likely because they are non-proliferating cells), and can regrow the tumour with classical cell hierarchy. Furthermore, a relatively quiescent, endogenous subset of cells that share properties with glioma stem cells and are resistant to TMZ, can regrow the tumour with classical cell hierarchy [10]. TMZ is an alkylating agent that targets proliferating cells and is currently the frontline in treatment protocols for GBM patients.

As noted above, rare quiescent sox2(+) stem cells were identified in a model of medulloblastoma [23]. The challenge for novel therapeutic strategies is the question of how to target quiescent non-dividing stem cells that are resistant to classical chemo and radiation therapies?

## 5. The Elephant in the Room: Targeting Quiescent Glioma Stem Cells (GSCs) and High-Grade Glioma (HGG) Cell Lines

GSCs, in contrast to brain tumour-propagating cells [27] and brain tumour-initiating cells [19,31], can be perpetuated and expanded serum-free [32] in vitro as high-grade glioma (HGG) cell lines [33,34,35], which are derived from patient biopsies. When grafted as xenografts in the brains of immune-compromised mice, HGG cell lines form orthotopic, intracranial tumours that retain their invasive potential and recapitulate much of the pathology of the original tumour [18,19,27,31,32,34]. For this reason, it is considered that HGG cell lines represent GSCs. However, the behaviour and responses of HGG cell lines in cell culture might not necessarily correspond to intra-tumoural responses of glioma stem cells [30].

## 6. Expression of Calcitonin (CT) Receptor, a G Protein-Coupled Receptor, in HGG Cell Lines

CT Receptors are widely expressed through the lifecycle of organisms, in several physiological conditions and diseases, including several cancers such as GBM and prostate cancer [5,36]. At least two different isoforms (insert-negative CT_a_ Receptor and insert-positive CT_b_ Receptor) have been identified in mammals. The latter isoform appears to be expressed across mammalian species, except *Muroidea,* and has an extra 16–18 amino acid-insert at the beginning of the second transmembrane span. The relative pharmacology of these isoforms has been summarised [5] and discussed in detail [37,38,39]. CT Receptors are encoded by the *CALCR* gene, which has been mapped to human chromosome 7 q21.3. This region is frequently amplified in GBM [40,41]. Of note, *CALCR* is upregulated by the transcription factor Sp1 [42], which is inhibited by mithramycin, an anti-neoplastic factor used in a mouse model of medulloblastoma [23].

From a collection of 12 HGG cell lines [34], 5 (42%) expressed CT Receptor when cultured in vitro [43]. What percentage express CT Receptor in orthotopic mouse models is unknown. Pharmacological assessment of the status of the CT receptor in four of these five HGG cell lines demonstrated that only in one line, SB2b, was modest coupling of adenylyl cyclase observed, with no activation of other signalling pathways, namely, ERK1/2, p38 MAP kinases, or calcium mobilisation [44]. In view of the negative CT pharmacological profile, it is unlikely that CT analogues would prove useful for treating CT Receptor-positive GBM [44]. The negative profile raises the possibility that the CT Receptor is mutated in these HGG lines (PK1, JK2, and WK1), which would suggest a role for the CT Receptor as a tumour suppressor or that there is predominant gene expression of the insert-positive CT_b_ Receptor isoform [5] with a potential role as an oncogene.

## 7. An Immunotoxin That Binds CT Receptor

Our group, together with colleagues in Berlin, have developed an anti-CT Receptor immunotoxin based on the antibody that binds the extracellular epitope (mAb2C4). The potency of mAb2C4:RIPs (ribosome-inactivating proteins: dianthin, gelonin) was compared to the antibody:drug conjugate (ADC) mAb2C4:MMAE (monomethyl auristatin E) with and without the endosomal escape enhancer triterpene glycoside SO1861 [43]. The results showed, in the presence of SO1861, that mAb2C4:RIPs (EC_50_ 10–20 pM) were 250–300 times more potent than the ADC (EC_50_ 2.5–6 nM) with the serum-dependent cell line U87MG and serum-independent HGG cell lines SB2b, JK2, and WK1, but PK1 was resistant. From analysis of immunoblots, PK1 showed decreased levels of CT Receptor associated with the membrane fraction [44].

These results indicate that even with pharmacological inactivation of the CT Receptor (mutations or expression of the isoform, CT_b_ Receptor), the immunotoxin is a potent reagent to promote cell death of HGG cell lines. It remains to be established if there exists a subpopulation of quiescent cells in these HGG lines and whether they can be effectively targeted with the immunotoxin.

The data with the immunotoxin discussed above reflect cell-based studies rather than animal or clinical trials, in which effectiveness of the immunotoxin is likely to be more complicated. The animal studies will provide potential toxicity, efficacy, and comparative data with conventional treatments such as TMZ. There are many small clinical trials underway, as reviewed recently [45].

## 8. Further Challenges of Solid Tumours, in Particular, Glioblastoma

Figure 1B shows regions of oedema associated with glioblastoma, which is a common feature in which the normal blood–brain barrier is non-existent and provides an explanation for the penetration of antibodies in brain tumours [6].

To improve tumour penetration, several strategies have been adopted and integrated, including construction of smaller antibody derivatives such as scFv or diabodies and reducing renal clearance with incorporation of PEG. Further improvement of targeting antibodies and derivatives has been demonstrated for the pegylated diabody (Avibody) PEG-AVP0458, that targets the tumour-associated glycoprotein TAG-72 of ovarian and prostate cancers [46]. High tumour/blood ratios, a favourable bio-distribution (high tumour penetration with minimal organ uptake), and prolonged T_1/2_ were reported.

### 8.1. Nanobodies

Single-domain antibodies are antibody fragments from camelids or cartilaginous fish that have a single variable region of a heavy chain (VHH or VNAR, respectively) mounted onto a constant domain framework. VHHs have been of interest as both therapeutics [47] and research tools due to their small size (15 kDa, which favours penetration of solid tumours), high antigen binding affinity, and their increased stability across temperature and pH ranges. VHHs that recognise the target of interest are typically generated in a few steps, including immunisation of alpacas, cloning of the VHH repertoire from isolated plasma cells ligated into phage display vectors, to establish a single-domain library, and selections to obtain antigen-specific VHHs. Soluble VHHs can be expressed in the periplasm of *Escherichia coli* and extracted using an osmotic shock protocol. Humanised versions have been generated on a single-nanobody scaffold [48].

### 8.2. Designed Ankyrin Repeat Proteins (DARPins)

Designed ankyrin repeat proteins (DARPin^®^) [49,50] provide an exciting prospect for therapy and are among the promising non-immunoglobulin binding proteins that demonstrated high affinity and specificity for targets, as well as high biophysical stability. As DARPins are small proteins and since engineering of DARPins can be facilitated by thiol or click chemistry, a PEG molecule can be introduced at either the N- or C-terminus. This increases the hydrodynamic radius of the resulting DARPin beyond the size limit for renal clearance, which increases the serum half-life. The high affinity of DARPins for targets and small size are important for tumour accumulation and presumably effective penetration. The smaller size might also be important to avoid or minimise an immune response. Avoiding immune responses will need to be factored into strategies for tumour targeting and clinical trials, as such events will eventually compromise the efficacy of the therapy.

## 9. Conclusions

Patients diagnosed with glioblastoma have a poor prognosis and quality of life. Repeated cycles of therapy eventually induce resistance and induce cellular changes with altered genetic profiles as the tumour regrows. Targeting glioma stem cells that might have previously been quiescent is proposed as a strategy to combat minimal residual disease and patient relapse. In glioblastoma, the CT Receptor is found expressed by a high proportion of patient biopsies and is expressed by malignant glioma cells and putative glioma stem cells. Forty-two percent (42%) of high-grade glioma (HGG; representative of GSCs) cell lines, available from one source and isolated from patient biopsies, express the CT Receptor protein, which is pharmacologically inactive. An immunotoxin was developed that binds an extracellular epitope of the CT Receptor, is accumulated into HGG cell lines, and is a potent effector of cell death. Further improvements in the binding moiety to reduce size and increase penetration are discussed, with examples such as diabodies, nanobodies, DARPins, and pegylation.

## Figures and Tables

**Figure 1 cells-10-02347-f001:**
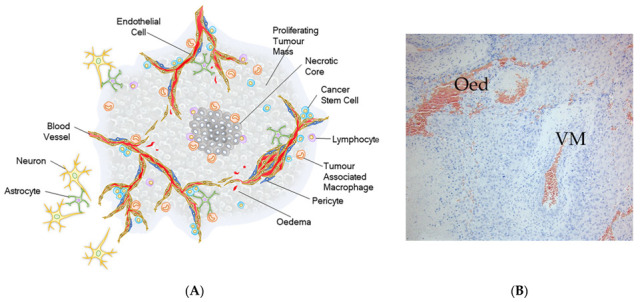
(**A**) Illustration of the major anatomical features of GBM [5]. (**B**) Haematoxylin & eosin stain of a tissue section of GBM with an example of vascular mimicry (VM) and oedema (Oed).

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
