# Peer review of "Strategic Development of an Immunotoxin for the Treatment of Glioblastoma and Other Tumours Expressing the Calcitonin Receptor"

_cells, 2021, doi:10.3390/cells10092347_

Round 1
Reviewer 1 Report
Title: Strategic development of an immunotoxin for the treatment of glioblastoma and other tumors expressing the calcitonin receptor
In this mini review, the authors well-described the hierarchy and complexity of GBMs and the current hurdle of improving the disease treatment. By introducing the high expression level of CT Receptor in patient biopsies and in nearly half of the established high-grade glioma cell lines, they proposed and discussed the therapeutic potential of targeting of CSCs expressing this receptor with pharmacological CT-response agents. This immunotoxin approach for GBM treatment is intriguing and has great potential for clinical use, and review of this could potentially broaden the knowledge in GBM field. However, the outline of this review is not well-designed and lacks of some background information even though it’s “mini” review and has words limit. But I believe it’s important and necessary to include those backgrounds which would benefit the understanding for wide-range readers. So generally I would suggest to reshape this review and could possibly accept it for publication in our journal with the modification.
- The author demonstrated the tumor hierarchy of GBM and cancer stem cells, but it lacks the connection of those two concepts, as what role cancer stem cells play in the GBM hierarchy and how the current regimens for GBM treatments fail to target all cancer stem cells.
- Check with the literature with more GBM stem cell markers, like CD133, etc.
- Discuss the difference of quiescent tumor stem cells with cancer stem cells in GBM, and how the pharmacological CT-response agents could target the quiescent population in order to improve the clinical outcome.
- It’s not clear that the calcitonin receptor is more in proliferating population or quiescent population in those high-grade glioma cell lines.
- Suggest to include more backgrounds of calcitonin receptor and it’s role in other soild tumors.
- The superior role of CT-response agents in GBM treatment is not clear illustrated, the authors could add a table to compare it with current standard-of-care TMZ agent.
Reviewer 2 Report
The review is too superficial, I understand that it is a mini review but it is necessary to give more information for example an in-depth study on cancer stem cell and calcitonin receptor (localization, quantization, function, etc.). Minor: in my opinion putting references on the titles is not easy for the reader to understand, if the references are essential to insert an introductory sentence to cite the references.
Round 2
Reviewer 1 Report
The authors have addressed most of my concerns. Only some minor issues need to be addressed before the acceptance.
- (a) The author demonstrated the tumor hierarchy of GBM and cancer stem cells, but it lacks the connection of those two concepts, as what role cancer stem cells play in the GBM hierarchy
(a) A connecting sentence has been inserted at line 66:
‘Currently, there are compelling data from animal models that GSCs play roles in tumourogenesis, tumour expansion and re-establishment of the tumour hierarchy following treatments. Small populations of GSCs have been identified that are quiescent and are resistant to conventional therapies that target dividing cells. These GSCs therefore offer a new potential target for therapies.’
- add relevant references to this statement
and
(b) how the current regimens for GBM treatments fail to target all cancer stem cells.
(b) This is dealt with in response to point 3) below and is inserted in line 88.
- the author only stated the sox2+ subpopulation can survive from the anti-mitotic treatment, which results in tumor recurrency. They should include the clarification of the shortage of conventional surgical removal, radiotherapy and TMZ treatment.
Reviewer 2 Report
The authors added new detailed informations in line with reviewer suggestion.
Author Response
Reviewer #2 made no further comments